# LOTTERY IMAGE PRIOR

## ABSTRACT

Deep Neural Networks (DNNs), either pre-trained (e.g., GAN generator) or untrained (e.g., deep image prior), could act as overparameterized image priors that help solve various image inverse problems. Since traditional image priors have much fewer parameters, those DNN-based priors naturally invite the curious question: *do they really have to be heavily parameterized?* Drawing inspirations from the recently prosperous research on lottery ticket hypothesis (LTH), we conjecture and study a novel "lottery image prior" (**LIP**), stated as: *given an (untrained or trained) DNN-based image prior, it will have a sparse subnetwork that can be trained in isolation, to match the original DNN's performance when being applied as a prior to various image inverse problems.* We conduct extensive experiments in two representative settings: (i) image restoration with the deep image prior, using an untrained DNN; and (ii) compressive sensing image reconstruction, using a pre-trained GAN generator. Our results validate the prevailing existence of LIP, and that it can be found by iterative magnitude pruning (IMP) with surrogate tasks. Specifically, we can successfully locate the LIP subnetworks at the sparsity range of 20%-86.58% in setting i; and those at sparsity range of 5%-36% in setting ii. Those LIP subnetworks also possess high transferability. To our best knowledge, this is the first time that LTH is demonstrated to be relevant in the context of inverse problems or image priors, and such compact DNN-based priors may potentially contribute to practical efficiency. Code will be publicly available.

## 1 INTRODUCTION

**Background** Deep neural networks (DNNs), in particular convolutional neural networks (CNNs), have been powerful tools for solving various image inverse problems such as denoising (Zhang et al., 2017; Guo et al., 2019; Lehtinen et al., 2018), inpainting (Pathak et al., 2016; Yu et al., 2018; 2019b), and super resolution (Ledig et al., 2017; Lim et al., 2017; Zhang et al., 2018). Conventional wisdom believes that is owing to DNNs' universal approximation ability and learning from massive training data. Yet, recent studies have revealed the specific architectures of CNNs have the inductive bias to represent and generate natural images well, and such favorable architecture inductive bias can work *independently from fitting specific training sets* (Ulyanov et al., 2018; Cheng et al., 2019; Bora et al., 2017; Heckel & Hand, 2019; Jalal et al., 2020).

For example, deep image prior (DIP) (Ulyanov et al., 2018) shows an **untrained** neural network can be used as a handcrafted prior that transfers well across multiple inverse problems. The authors attributed the success to the CNN architecture itself, that appeared to possess high noise impedance even with only random initializations. As another example, in compressive sensing, Bora et al. (2017); Jalal et al. (2020) replaced the common structural assumptions such as sparsity with a **pre-trained** generative adversarial networks (GAN). The underlying rationale lies in that a pre-trained generator should (approximately) represent the notion of a vector being or more likely in the target domain such as natural images; in other words, a sample more like a natural image will be closer to the output range of the pre-trained generator. In this paper, we refer to such general-purpose image prior parameterized by (either *untrained* or *pre-trained*) DNNs as *DNN-based image priors*.

Recall that, classical image regularizers in the spatial or frequency domains are often not learning-based (Tomasi & Manduchi, 1998; Sardy et al., 2001; Dabov et al., 2007), or rely on compact learning models (Cao et al., 2008; Elad & Aharon, 2006; He et al., 2015). On contrary, DNN-based image priors have a massive number of parameters (we compare the parameter numbers between the full model and the sparse subnetwork in Table. 2), typically magnitudes more than the image (even the image size) size. The two extremes invite the natural question: *Can we identify highly compact*

Figure 1: Overview of our work. In both untrained and pretrained DNN-based image priors, we study the existence of *Lottery Image Prior* (LIP) that could transfer to various image inverse problems such as denoising, inpainting, super-resolution, and/or compressive sensing restoration.

*DNN-based image priors, that are same effective?* We note that, diving into this question has two-fold appeals. On the algorithmic side, that could help us understand further how the topology and connectivity of CNN architecture itself will affect the effectiveness of those priors, and to what extent sparsity could be relevant. On the practical side, if we could provide an affirmative answer to this question, that would potentially lead to more computationally savings when applying those DNN-based priors in practice, leading to faster restoration or computational imaging with them.

Towards the above question, the tool we refer to in this paper is the recently emerged *Lottery Ticket Hypothesis* (LTH) (Frankle & Carbin, 2018; Frankle et al., 2020a). LTH suggests that every dense DNN has an extremely sparse "matching subnetwork", that can be trained in isolation to match the original dense DNN's accuracy. While the vanilla LTH studies training from random scratch, the latest works also extend similar findings to fine-tuning the pre-trained models (Chen et al., 2020a; 2021a). LTH has widespread success in image classification, language modeling, reinforcement learning and multi-modal learning, e.g., (Yu et al., 2019a; Renda et al., 2020; Chen et al., 2020a; Gan et al., 2021).

**Our Contributions** Drawing inspirations from the LTH literature, we conjecture and empirically study a novel "lottery image prior" (**LIP**), stated as:

*Given an (untrained or trained) DNN-based image prior, it will have a sparse subnetwork that can be trained in isolation, to match the original DNN's performance when being applied as a prior to regularizing various image inverse problems.*

Studying this new problem is, however, **NOT** a naive extension from the existing LTH methods, owing to several technical barriers: **(a)** till now, LTH has not been demonstrated for image inverse problem or DNN-based priors, to our best knowledge. Most LTH works studied discriminative tasks, with one exception (Chen et al., 2021c). It is therefore uncertain whether high-sparsity DNN is still viable for reconstruction-oriented tasks; **(b)** existing LTH works typically require a full training set to locate the sparse subnetwork mask, whereas our LIP settings are only not data-rich. For example, DIP needs the DNN to be trained to overfit one specific image, making it drastically different from previous problems; **(c)** the objectives between finding the sparse mask (e.g., learning the prior) and fitting the sparse subnetwork (e.g., using the prior) are often unaligned in LIP problems. For example, DIP will overfit a corrupted input image (during which the sparse mask will be found) in order to reconstruct a clean output image (when the found sparse subnetwork will be used); the pre-trained generator will also be used towards a different goal (regularizing compressive sensing) from their original pre-training task (generating realistic images).

Our extensive experimental study confirms the existence of LIP in two representative settings: **(i)** image restoration with the deep image prior, using an untrained DNN (as shown in Fig. 2); and **(ii)** compressive sensing image reconstruction, using a pre-trained GAN generator. Using iterative magnitude pruning (IMP) with surrogate tasks (the overview of our work paradigm is in Fig. 1), we can successfully locate the LIP subnetworks at the sparsity range of 20%-86.58% in setting i; and those at the sparsity range of 5%-36% in setting ii. Those LIP subnetworks also possess high transferability. For example, the LIP ticket found in the setting i transfer well across not only different images, but also different tasks such as denoising, inpainting and super-resolution. Our contributions are summarized below:

- The first comprehensive study on LTH in DNN-based image priors and inverse problems, establishing the "lottery image prior" (**LIP**) and demonstrating the prevailing relevance of LTH more broadly than previously typical settings.

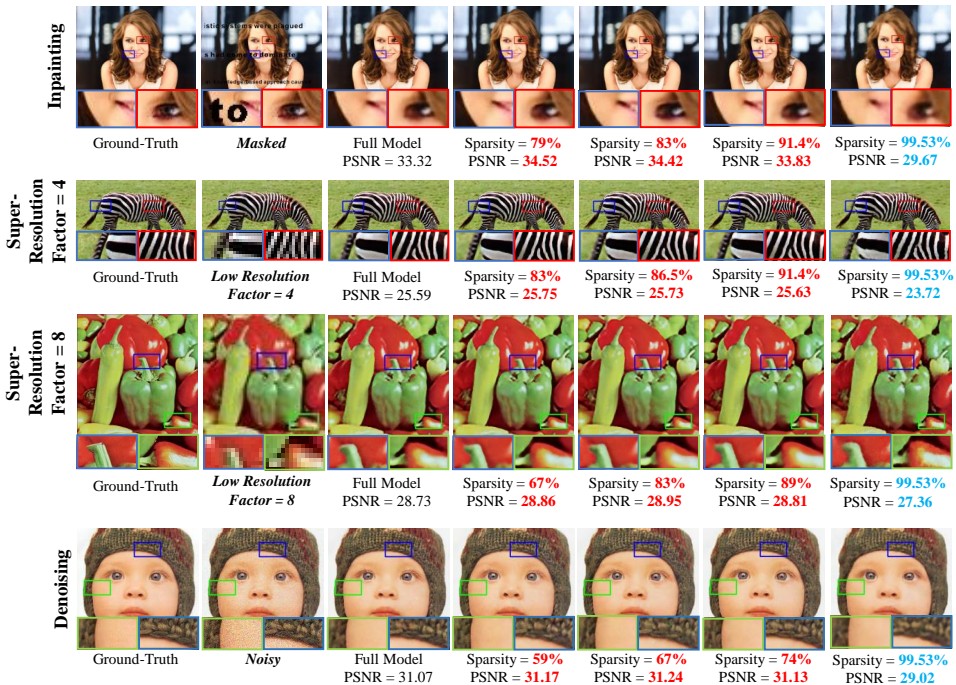

Figure 2: **LIP** visual results: inpainting (row 1), super-resolution (rows 2/3) and denoising (row 4). The last column (in blue) intends to display the results with the most extremely sparse subnetwork.

- The investigation of both untrained and pre-trained DNNs as image priors, verifying that LIP subnetworks can be found in both settings, by overcoming several impediments such as severely limited training data and task objective mismatch.
- High transferability of LIP subnetworks across data/datasets, and various inverse problem tasks. Our finding reflects the underlying common image prior that is agnostic to specific data or task, through the lens of the CNN architecture together with sparsity.

## 2 BACKGROUND WORK

**Lottery Ticket Hypothesis**  LTH (Frankle & Carbin, 2018) states that the dense, randomly initialized DNN contains a sparse matching subnetwork, which could reach the comparable or even better performance by independently being trained for the same epoch number as the full network do. Since then, the statement has been verified in a variety of fields, such as image classification (Frankle & Carbin, 2018; Liu et al., 2018; Wang et al., 2020; Evci et al., 2019; Frankle et al., 2020b; Savarese et al., 2019; Yin et al., 2019; You et al., 2019; Ma et al., 2021; Chen et al., 2021a), natural language processing (Gale et al., 2019; Chen et al., 2020a), reinforcement learning (Yu et al., 2019a), lifelong learning (Chen et al., 2020b), graph neural networks (Chen et al., 2021b), and adversarial robustness (Cosentino et al., 2019). Rewinding was proposed by (Frankle et al., 2019) to scale up the LTH to large models and datasets. The found matching subnetworks also demonstrate transferability across datasets and tasks (Morcos et al., 2019; Desai et al., 2019).

**Deep Image Prior and Its Variants**  Despite CNN's tremendous success on various imaging tasks, their outstanding performance is often attributed to massive data-driven learning. DIP (Ulyanov et al., 2018) pioneered to show that CNN architecture alone has captured important natural image priors: by over-fitting a randomly initialized untrained CNN to a single degraded image (plus some early stopping), it can restore the clean output without accessing ground truth. Follow-up work (Mataev et al., 2019) strengths DIP performance by incorporating it with the regularization by denoising (RED) framework and a series of works (Mastan & Raman, 2020; 2021) use the contextual feature learning method to achieve the same goal of DIP. Besides natural image restoration, DIP was successfully applied to PET image reconstruction (Gong et al., 2018), dynamic magnetic resonance imaging (Jin et al., 2019), unsupervised image decomposition (Gandelsman et al., 2019) and quantitative phase imaging (Yang et al., 2021). Heckel & Hand (2018) further demonstrated that even an under-parameterized non-convolutional model, named "Deep Decoder", can over-fit a sin-

gle degraded image like DIP did, which does not critically rely on early stopping or so. Chen et al. (2020c) was the first to study the possibility to optimize CNN architectures for capturing stronger image priors in DIP, via leveraging Neural Architecture Search (NAS).

**Compressive Sensing using Generative Models**   Compressive Sensing (CS) reconstructs an unknown vector from under-sampled linear measurements of its entries (Foucart & Rauhut, 2013), by assuming the unknown vector to admit certain structural priors. The most common structural prior is to assume that the vector is $k$-sparse in some known bases (Candes et al., 2006; Donoho, 2006), and more sophisticated statistical assumptions were also considered (Baraniuk et al., 2010). However, those priors are inevitably oversimplified to depict the high-dimensional manifold of natural images. Bora et al. (2017) presented the first algorithm that used pre-trained generative models such as GANs, as the prior for compressed sensing. As a prior, the pre-trained generator encourages CS to produce vectors close to its output distribution, which approximates its training image distribution. Significant research has since followed to better understand the behaviours and theoretical limits of CS using generative priors, e.g., (Hand & Voroninski, 2018; Bora et al., 2018; Hand et al., 2018; Kamath et al., 2019; Liu & Scarlett, 2020; Jalal et al., 2020).

## 3   PRELIMINARIES AND APPROACH

### 3.1   NEURAL NETWORK AS PRIORS: TWO SETTINGS

Although the implementation process of the GAN Compressed Sensing (CS) task and the DIP restoration task are different, yet we combine the two studies in this one paper for two reasons: 1) we are motivated by two current LTH streams: networks with weights trained from scratch (Frankle & Carbin, 2018) and networks with pre-trained weights (Chen et al., 2021a). 2) Network structures with random weights or pre-trained weights can be priors, which is the commodity we view as the two could be unified. In **setting i**, we use an untrained CNN as the dense model, and solve the DIP optimization (Ulyanov et al., 2018)

$$\theta^* = \arg\min_\theta E(f_\theta(z); \tilde{x}), \ x^* = f_{\theta^*}(z), \tag{1}$$

where $E(;)$ denotes the Mean Square Error (MSE), $\tilde{x}$ is the corrupted version of the image $x \in \mathbb{R}^{3 \times H \times W}$, $f_\theta$ represents the dense model $f$ with initial parameter $\theta$ and $z$ is the random tensor that $z \in \mathbb{R}^{C \times H \times W}$. We choose the same hourglass architecture with skip connections as in (Ulyanov et al., 2018), to be the dense model by default.

In **setting ii**, we follow Bora et al. (2017) to use a pre-trained GAN generator, to reconstruct the unknown vector $x^* \in \mathbb{R}^n$, after observing $m < n$ linear measurements of its entries with noise: $y = Ax^* + \eta$, where $A \in \mathbb{R}^{m \times n}$ is the measurement matrix and $\eta \in \mathbb{R}^m$ is the noise. Within the range of the pre-trained generator prior, GANs could reconstruct the vector $x^*$ with high perceptual quality. Following the recommendation of Jalal et al. (2020), we use the official pre-trained model PGAN (Karras et al., 2017).

### 3.2   FINDING LOTTERY TICKETS

**Datasets**   In **setting i**, we use the popular Set5 (Bevilacqua et al., 2012) and Set14 (Zeyde et al., 2010) datasets. Besides, we evaluate the transferability of subnetworks on image classification datasets such as MNIST (LeCun et al., 2010) and CIFAR10 (Krizhevsky et al., 2009). In **setting ii**, our evaluation dataset is CelebA-HQ (Lee et al., 2020; Karras et al., 2017), following Jalal et al. (2020).

**Subnetworks**   Consider a network $f(x; \theta)$ parameterized by $\theta$ with input $x$, then a subnetwork is defined as $f(x; m \odot \theta)$, where $\odot \in \{0,1\}^d$, $d = ||\theta||_0$ and $\odot$ is the element-wise product. Let $\mathcal{A}_t^{\mathcal{T}}(f(x; \theta))$ to be the training algorithm, that is, training model $f(x; \theta)$ on the specific task $\mathcal{T}$ with $t$ iterations. We also denote the random initialization weight as $\theta_0$ and the pre-trained weight as $\theta_p$; $\theta_i$ as weight at the $i$-th training iteration and $\mathcal{E}^{\mathcal{T}}(f(x; \theta))$ the model performance evaluation.

**Finding Subnetworks**   Following the definitions of Frankle et al. (2020a), we define that if the subnetworks is *matching*, it satisfies the following conditions (we use $\theta_p$ for example, to denote a pre-trained lottery ticket (Chen et al., 2020a; 2021a); $\theta_0$ can be defined likewise):

$$\mathcal{E}^{\mathcal{T}}(\mathcal{A}_t^{\mathcal{T}}(f(x; \theta_p))) \leq \mathcal{E}^{\mathcal{T}}(\mathcal{A}_t^{\mathcal{T}}(f(x; m \odot \theta))). \tag{2}$$

That is a matching subnetwork that performs *no worse* than the dense model under the same training algorithm $\mathcal{A}^{\mathcal{T}}$ and the evaluation metric $\mathcal{E}^{\mathcal{T}}$. Similarly, we define the *winning tickets*: if a *matching* subnetwork $f(x; m \odot \theta)$ has $\theta = \theta_p$, then it is the *winning tickets* under the training algorithm $\mathcal{A}^{\mathcal{T}}$.

In practice, we find the matching subnetworks in the following ways: **1)** IMP on the dense model with only one image (applicable to the image restoration task on the same domain of images); **2)** IMP on the dense model with various kinds of images through weight-sharing (applicable to the image restoration tasks on images from different domains); **3)** LTH IMP on the dense model with the sub-dataset (applicable to the CS task with pre-trained GANs).

**Ticket Finding Objectives**  When finding LIP subnetwork in untrained DNN, i.e., DIP, there is a problem that may be easily overlooked: *what is the target during IMP training?* Specifically, since the training target in Eq. 1 for DIP is $\tilde{x}$ (the corrupted image), the model parameters may easily overfit the corrupted image during IMP training if we do not fine-tune the training epochs. Then the obtained mask will contain the information of corrupted images instead of the desired image prior (but we can also find the effective subnetworks by noisy targets, experiments are summarized in Table. 3 and 4 in Supplementary Materials). Therefore, we modify the optimization objectives of DIP during IMP training: $\arg\min_\theta E(f_\theta(z); x)$ to ensure that model parameters learn the clean image prior.

**Evaluations of Subnetworks**  After obtaining the matching subnetwork $f(x; m \odot \theta)$, we evaluate their performances by 1) resetting the model parameter $\theta$ to initialization weight $\theta_0$' 2) adding the mask $m$ to the model; 3) training the model to another $N$ iterations. We evaluate the DIP subnetworks performances mainly through Peak Signal to Noise Ratio (PSNR) values, plus reconstruction errors (defined in (Jalal et al., 2020) for compressive sensing with GANs.

**Pruning Methods**  We use the *standard pruning* method (Han et al., 2015), which iteratively prune the 20% of the model weight each time. For **setting i**, our basic algorithm performs IMP over just one single image (i.e., DIP's default setting), and the algorithm is summarized in Algorithm 1 (Appendix). We further design an extended algorithm, that can perform IMP for DIP over multiple images, through backbone weight sharing: the algorithm is outlined in Algorithm 2 (Appendix). We will discuss the algorithm variants in Section 4. For **setting ii**, we use IMP to find winning tickets in *pre-trained GANs*, following the routine in (Chen et al., 2021c). In each iteration of IMP, we will first fine-tune the pre-trained GANs on a (sub-)dataset, prune 20% of the remaining weights, and reset to the pre-trained weights. Note that we also include other pruning methods to compare the effectiveness of the *matching* subnetworks such as random pruning (*randomly generate the sparsity mask $m'$*), and pruning-at-initialization methods, e.g., SNIP (Lee et al., 2018).

## 4    LIP FOR DEEP IMAGE PRIOR WITH UNTRAINED DNNS

In this section, we will investigate the lottery image prior (LIP) property under **setting i**, for deep image prior (DIP) models. Model performance is measured by PSNR (*Peak Signal to Noise Ratio*) between the restored images and the clean ground truths (SSIM (*Structural Similarity*) results are in supplementary materials). We run all experiments with three different random seeds. All images used for plotting results are summarized in Fig. 14.

In DIP, for each degraded image to be restored, an untrained DNN will be over-fitted over that single image (with proper early stopping). This casts a crucial difference from typical LTH settings where winning tickets are identified using IMP over a distribution of data and are verified to be effective when trained on data from that specific distribution. Therefore, besides the usual properties in winning tickets such as the existence, the superiority over other pruning methods, and the effect of rewinding, we will strive to answer two more questions about LIP in DIP. The first question is: *can we extend the IMP algorithm for single-image DIP to multi-image and thus improve the performance of winning tickets hopefully?* The second question is *to what extend the LIP found in DIP setting can be transferred?* Answering these two questions not only provides a clearer picture of LIP, but also hints a new perspective for a deeper understanding of LTH.

**Existence of LIPs**  In DIP setting, we first find the winning tickets with LIP property by implementing the single-image IMP on the hourglass model (i.e., the DIP model used in Ulyanov et al. (2018). We adopt the modified objectives using clean images as labels as we discussed in Sec. 3.2. The algorithm is described in detail in Algorithm 1. We apply the implemented algorithm on Set5

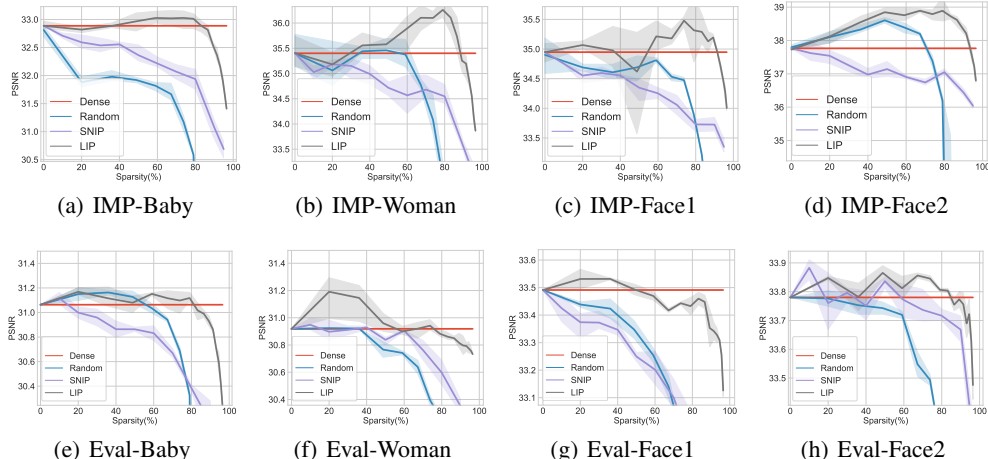

|     |     |     |     |
|-----|-----|-----|-----|
| (a) IMP-Baby | (b) IMP-Woman | (c) IMP-Face1 | (d) IMP-Face2 |
| (e) Eval-Baby | (f) Eval-Woman | (g) Eval-Face1 | (h) Eval-Face2 |

Figure 3: Experimental results of finding LIP in **setting i** (i.e., DIP). The first row of the figure summarizes the LTH IMP training loops and the second row denotes the evaluation of found LIP. Note that we compare the LTH IMP with Random Prune (Random) and SNIP (Lee et al., 2018) prune methods, on images from different (Baby and Woman) or same domains (Face1 and Face2).

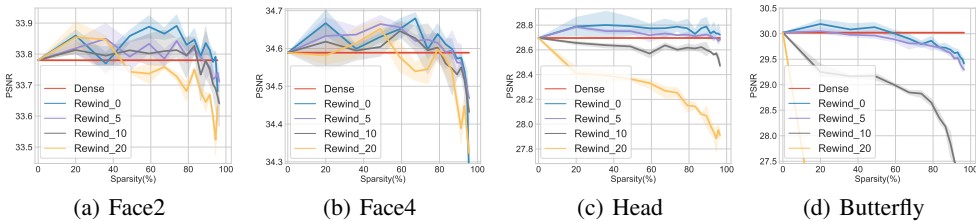

|     |     |     |     |
|-----|-----|-----|-----|
| (a) Face2 | (b) Face4 | (c) Head | (d) Butterfly |

Figure 4: Experiments of the rewind strategy (background task: denoising). Note that we train the model with $N$ epochs in IMP and Rewind_j means rewinding the ticket parameter to $\theta_j$, the weights after $j\% \times N$ steps of training.

(Bevilacqua et al., 2012) and Set14 (Zeyde et al., 2010) datasets to obtain the sparse subnetworks[1] and evaluate these subnetworks on *the denoising task*.

Results of single-image IMP in Fig. 3 (curves in red colors) verify that the LIP exists in the DIP setting. To be specific, during the IMP finding process when we use clean images in DIP objectives, we are able to find the winning tickets with LIP on untrained DNN at sparsity as high as 86.58%. While at the evaluation stage, when we only have access to the degraded images, we find that the winning tickets found with the modified objectives are still applicable, matching the dense performance at sparsity as high as 83.23%.

Besides the dense model baselines, we also compare the single-image IMP with *Random Pruning*, and a pruning-at-initialization method *SNIP* (Lee et al., 2018), whose results are also presented in Fig. 3 in other colors. Specifically, we clearly observe from the first row in Fig. 3 that the single-image IMP for DIP outperforms random pruning and SNIP at a wide sparsity range [20%, 96%]. Interestingly, we find random prune is a good competitor to SNIP sometimes, but both of them suffer significant performance decrease at extreme sparsities (over 80%), where IMP still persists.

**The Effect of Weight Rewinding** In this part, we study the effect of weight rewinding (Frankle et al., 2019) when applied to the single-image IMP for DIP models. Weight rewinding is proposed to scale LTH up to large models and datasets. Specifically, we say we use $p\%$ weight rewinding if we reset the model weights at the end of each IMP iteration to the weights in the dense model after a $p\%$ ratio of training steps within a standard full training process, instead of the model's random initialization. For the single-image IMP in DIP, we consider 5%, 10% and 20% weight rewinding schemes. The resulting models are denoted as *Rewind_5*, *Rewind_10* and *Rewind_20*, respectively. The results of different weight rewinding schemes are summarized in Fig. 4. We can see that weight

---

[1]We prune 20% of the remaining weights in each IMP iteration, resulting in sparsity ratios $s_i = 1 - 0.8^i$.

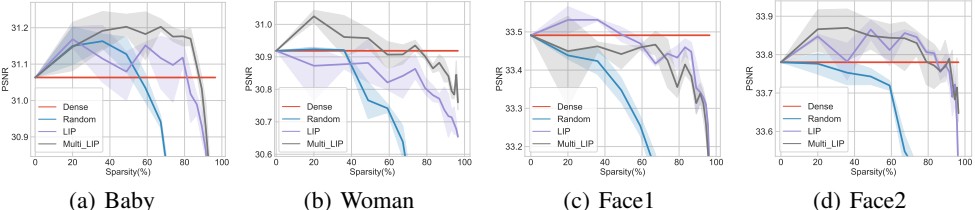

|  (a) Baby  |  (b) Woman  |  (c) Face1  |  (d) Face2  |

Figure 6: Experimental Results of Multi LIP on images from same/different domains. We compare Multi LIP with LIP and random prune methods. The background task is denoising in setting i.

rewinding is not beneficial for identifying LIP in the DIP setting. Too much rewinding (10% and 20%) even hurts performance or fails it completely. We conjecture that this is due to the extremely low data complexity in DIP (single image).

**Is Multi-image IMP A Good Extension in DIP?** DIP over-fits an untrained DNN on a single image. A reasonable conjecture is that the DNN will be dedicated to the features in that specific image and so should the resulting winning ticket be if we apply single-image IMP to the DIP model. We then ask that whether we can co-consider multiple images during IMP for DIP models to find winning tickets that are suitable for more general features and thus yield better image restoration performance? To verify our idea, we propose a new *multi-image IMP* for DIP where we replace the single-image DIP objective with the average of multiple images during the IMP process. Note that all images will share the same fixed random code during IMP. Otherwise, different random codes will intervene with one another because they lead to different winning tickets, resulting in a share subnetwork with inferior performance. The algorithm is formally described in Algorithm 2.

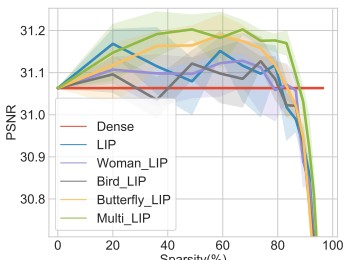

Figure 5: We evaluate different single-image LIPs and the multi-image LIP on Baby image to measure LIP's transferability (cross images).

We evaluate the multi-image IMP in two different settings: (i) *cross-domain* setting where we apply the multi-image IMP to the five images from Set5 (Bevilacqua et al., 2012); (2) *single-domain* setting where we apply the multi-image IMP to five images of human faces with glasses. We think images from Set 5 are from more diversified domains because they include bird, butterfly and human face contents. We compare single-image IMP winning tickets found on the Baby and the Woman images from Set5 with the cross-domain ticket, and the single-image IMP winning tickets found on Face-1 and Face-2 with the single-domain ticket. Results presented in Fig. 6 show that multi-image IMP significantly improves the quality of the winning tickets in the cross-domain setting, which is aligned with our previous hypothesis.

**To what extent can LIP tickets be transferred?** Transferability is an important metric to measure the usefulness of LIP in the DIP setting. It would significantly undermine the practical application of LIP if the winning tickets can only be dedicatedly found for each image. In this part, we evaluate the transferability of LIP for DIP models from three perspectives, i.e., the transferability *between images*, *between image restoration tasks* and *between high-level tasks*.

*Observation 1: LIP can transfer across images.* We identify the single-image LIPs for all image from Set5 and the multi-image LIP on Set5 using the new multi-image IMP proposed in the last part. Then we evaluate all the above LIPs on the Baby image, results being presented in Fig. 5. We can see that the LIPs found on other images, even those from the Bird and the Butterfly images, can perform comparably well with LIP dedicatedly found on the Baby image. This shows that LIP has reasonable transferability across images, even for those coming from slightly different domains (for example, the Baby and the Butterfly). We can also observe the fact that the multi-image LIP outperforms all single-image LIPs, as side support for the superiority of multi-image LIP in the cross-domain settings.

*Observation 2: LIP can transfer across image restoration tasks but not to other high-level tasks.* Ulyanov et al. (2018) showed the effectiveness of DIP in different image restoration tasks including denoising, inpainting and super-resolution. We conduct experiments to verify if a LIP winning ticket identified on one image restoration task can be *re-used* in another because they share some common

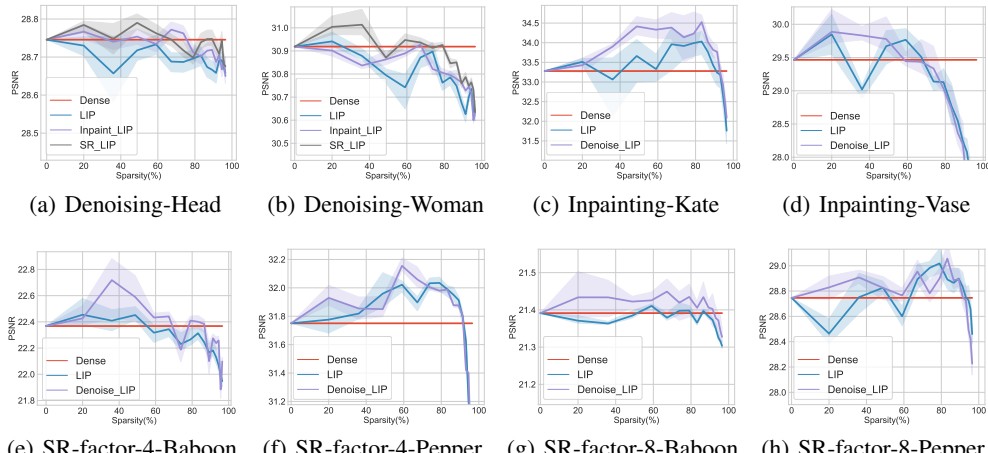

(a) Denoising-Head  (b) Denoising-Woman  (c) Inpainting-Kate  (d) Inpainting-Vase

(e) SR-factor-4-Baboon  (f) SR-factor-4-Pepper  (g) SR-factor-8-Baboon  (h) SR-factor-8-Pepper

Figure 7: Transferability (cross tasks) experimental results. We study the transferability of denoising LIP on the restoration tasks such as inpainting and super-resolution (SR); we also study the inpainting and SR LIP on the denoising task. We consider two SR scale factors = 4, 8.

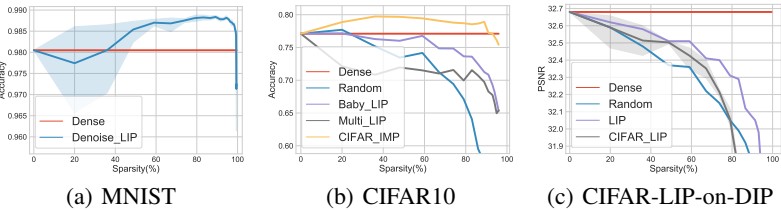

(a) MNIST  (b) CIFAR10  (c) CIFAR-LIP-on-DIP

Figure 8: Transferability experiments. We test the denoising LIP on MNIST and CIFAR10 datasets (left 2 figures). Note that we replace the last convolutional layer of DIP models with the linear layer and load the same initial weights. We evaluate the CIFAR LIP on the denoising task.

interests? Furthermore, if the above question has an affirmative answer, is such transferability sustainable when transferring to or from some high-level tasks such as classification?

To evaluate the transferability of LIP between image restoration tasks, we first find three LIPs for the denoising, inpainting and super-resolution tasks respectively and then transfer between them, as shown in Fig. 7. We observe that a LIP winning ticket transferred from another image restoration task always yields restoration performance comparable with the single-image LIP found on the original task, sometimes even better, for examples in Fig. 7(a) and 7(g).

We then evaluate the transferability of LIP between the denoising task and image classification task on CIFAR-10. We apply the standard IMP on the DIP model, i.e., the hourglass architecture in (Ulyanov et al., 2018), to obtain winning tickets on MNIST and CIFAR-10 datasets. We show the results of transferring the denoising LIP to MNIST in Fig. 8(a) and CIFAR-10 in 8(b) and the CIFAR-10 winning tickets to denoising task in Fig. 8(c). Although transferring denoising LIP to MNIST seems to yield winning tickets due to MNIST's low complexity, its transfer to CIFAR-10 is unsuccessful. Interestingly, the multi-image LIP becomes a worse ticket on CIFAR-10 than the single-image LIP. Moreover, transferring winning tickets on CIFAR-10 back to the denoising DIP task also fails to generate winning tickets that are comparable with denoising LIPs.

Based on those observations, we conjecture that *the architecture priors needed to "win the lottery ticket" may substantially differ between low-level (reconstruction, restoration,...) and high-level (classification, recognition,...) vision tasks, but seems to be quite overlapped/shareable among different tasks that are all low-level (or high-level).* The high-level task transferability of winning tickets has been found in prior works (Morcos et al., 2019; Desai et al., 2019; Chen et al., 2021a). We believe this interesting "incompatibility" between low-level and high-level tasks is an unstudied new direction, and leave this for our future work.

## 5 LIP FOR COMPRESSIVE SENSING WITH GENERATIVE MODELS

Finding LIP in GAN priors for compressive sensing task is more natural than in the **setting i**. The training process of GANs is more similar to a typical IMP setting where the models are trained on a distribution of data for image classification. And the existence of winning tickets in GANs for

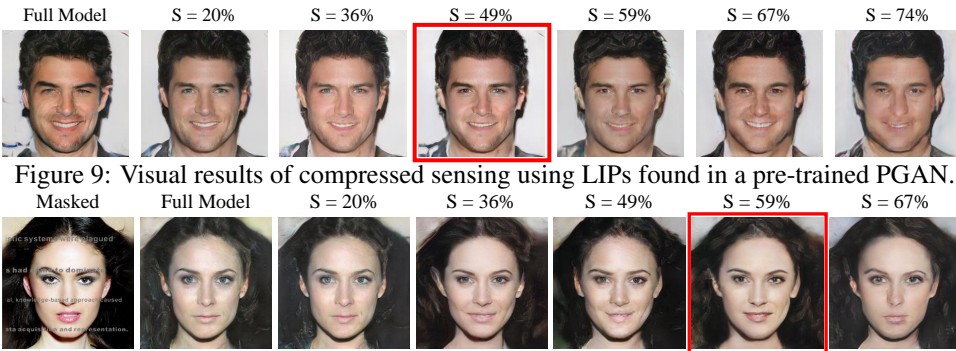

Figure 9: Visual results of compressed sensing using LIPs found in a pre-trained PGAN.

Figure 10: Visual results of inpainting using LIPs found in a pre-trained PGAN.

the generative task has been shown in (Chen et al., 2021c). But it has not been shown for GANs as image priors for the compressive sensing task. In this section, we first describe how we identify LIP in pre-trained GANs using IMP and show the existence of LIP for the compressive sensing task. Then we show that the LIPs found can be transferred to other image restoration tasks such as denoising and inpainting.

**Existence of LIP in GANs for Compressive Sensing**   We use PGAN (Karras et al., 2017) pre-trained on CelebA-HQ dataset (Lee et al., 2020) as the model in this section. To obtain LIP tickets in pre-trained GANs, we apply the IMP algorithm. In each IMP iteration, PGAN is first fine-tuned on 40% of images in celebA-HQ for 30 epochs, has 20% of its remaining weights pruned, and then reset to the pre-trained weights. We only prune the generator in the IMP process because it is found in Chen et al. (2021c) that pruning discriminator or not only has marginal influence on the quality of the winning tickets. We then evaluate the tickets on the compressive sensing task following the setting in Jalal et al. (2020): we fix the number of measurements to 1,000 with 20 corrupted measurements, and minimize the MOM objective for 1,500 iterations to recover the images. We compare the performance (measured in per-pixel reconstruction error) of LIP with the dense baselines in the first row of Table. 1 and provide a visual example in Fig. 9. Tickets with higher sparsities can match the reconstruction performance of the dense model, confirming the existence of the winning tickets.

| Sparsity | 0% | 20% | 36% | 49% | 59% | 67% | 74% |
|---|---|---|---|---|---|---|---|
| **Random-CS** | 0.0725 | 0.0963 | 0.1165 | 0.1276 | 0.2184 | 0.2086 | 0.3655 |
| **LIP-CS** | 0.0725 | 0.0744 | 0.0732 | 0.0737 | 0.0711 | 0.0728 | 0.0728 |
| **Random-I** | 0.0541 | 0.0682 | 0.0748 | 0.08101 | 0.1142 | 0.1904 | 0.2195 |
| **LIP-I** | 0.0541 | 0.0542 | 0.0504 | 0.0514 | 0.0506 | 0.0524 | 0.0509 |

Table 1: Experimental results of GAN LIP. We evaluate the LIPs found in PGAN on the compressed sensing (CS) and the inpainting (I) tasks. The results are based on celebA-HQ dataset (Lee et al., 2020). Note that we use the MSE (per pixel) to evaluate the LIP effectiveness and compare the LIP with random pruning results.

**Transfer to other image restoration tasks – inpainting**   Besides the experiments on the compressed sensing restoration tasks, we also evaluate the effectiveness of GAN LIP on the inpainting task: masking the image and then optimize the input tensor of generator in the GAN LIP range to reconstruct the pristine image. More formally, consider the input tensor $z' \in \mathbb{R}^{1 \times 512}$, the pristine image $x$ sampled from celebA-HQ, inpainting mask $A$ (binary mask), masked image $y = Ax$ and a generator $G$, then the optimization loss function is: $||AG(z) - y||_2$. The results are summarized in Table. 1 and Fig. 10, demonstrating the transferability of GAN LIP.

## 6   CONCLUSION

In this paper, we successfully find the lottery image prior (LIP) via *lottery ticket hypothesis* and we have also empirically demonstrated the prevailing existence of LIP in image inverse problems such as denoising, inpainting, super-resolution and compressed sensing restoration. Specifically, we show that subnetworks with high sparsity can still retain the beneficial image prior properties in both settings. We also prove the powerful transferability of LIP across these tasks, reflecting its promising application potential.

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

## A    SUPPLEMENTARY MATERIALS

In this section, we will present more experimental results of LIP subnetworks (e.g., the layer-wise sparsity ratio, discussions about the training targets, learning curves of different training targets, etc.) to better understand the *Lottery Image Prior*.

**Images Used in Experiments**    In Fig. 14, we organize and present the images used in this paper with their names. Note that these images are sampled from Set5 (Bevilacqua et al., 2012) and Set14 (Zeyde et al., 2010) datasets, and we use their default names.

**Algorithm**    To begin with, there exist two ways of finding **LIP**: 1) Single-image based IMP. 2) Multi-image based IMP (weight-sharing). We have formulated the 2 method in Algorithm. 1 and 2, respectively. The major difference lies in the loss function of IMP training. Specifically, for the denoising task, the loss function of single-image IMP is $E(f(z; \theta \odot m), x)$, where $z$ is a randomly initialized tensor, $x$ is the groundtruth image and $E(;)$ represents the MSE distance. In parallel, the weight-sharing IMP loss function is $\sum_{a=1}^{n} E(f(z; \theta \odot m), x_a)$, $a \in [1, n]$, where $n$ represents the number of shared weight images (e.g., in Set5 dataset, $n = 5$.). Note that we use the weight-sharing method to find the LIP with the powerful transferability on images from different domains and use the single-image IMP on the images from the same domains (such as human face images).

**Parameter Redundancy Problem**    We discover that the commonly used hourglass model in DIP (Ulyanov et al., 2018) is high parameter complex. The statistic results of parameter numbers is summarized in Table. 2. We compare the parameter numbers of dense DIP models with the found winning tickets and surprisingly discover that the winning tickets could perform better than the full model while containing 2 million parameters fewer. This phenomenon motivates us to suspect that there is a high possibility of finding the matching subnetworks of the pristine dense DIP model, which indicates that the subnetworks may also contain the outstanding image prior property as the dense one does.

|  | Dense | Tickets |
|---|---|---|
| PSNR | 30.35 | **30.61** |
| Parameters (non-zero) | 2.2M | **0.2M** |

Table 2: The comparison of parameter numbers in full model and the winning tickets. Note that we evaluate the PSNR values on the image Bird.

|  | Dense | Noisy M ($s = 50\%$) | Clean M ($s = 50\%$) |
|---|---|---|---|
| PSNR | 31.06 | 30.76 | **31.21** |

Table 3: Comparison results of different training targets. We train the DIP model on one image (Baby.png) for 6000 epochs and separately set the training target to $x$ (clean target) and $\tilde{x}$ (noisy target). Then we evaluate the masks on the same image for 3000-epoch denoising task.

**Training Target of Finding LIP**    When making the IMP training loop, one question arises: *How to iteratively prune the model while maintaining the image prior property in DNNs?* In this paper, we propose to set the training label of the DIP model to clean image $x$. Since the DIP model restore the corrupted image (e.g., the noisy image, masked image and low-resolution image) by training itself with the corrupted target $\tilde{x}$, and the best restoration image usually occurs at some points of the process not always the training ending (DIP model training often requires the researcher to design the training settings by hand). Therefore, if we set the IMP training target to $\tilde{x}$, the model parameters may overfit the corrupted image, which will degrade the desired image prior property. In Table. 3, we compare the ticket performances between the noisy target IMP trained (Noisy M) and clean target IMP trained (Clean M) at the same sparsity $s = 50\%$. This experiment is done on the Baby image and the background task is denoising. In order to test the effect of overfitting, we set the DIP training epochs to 6000 (3000 is usually used for the Baby image). We observe that the Clean M reaches 0.45 higher PSNR value than Noisy M and the Clean M performance is higher than the Dense model, which demonstrates the importance of choosing the training target during IMP.

However, the above observations do not indicate that our proposed method is applicable only to the ground-truth image settings. We used clean images as the training targets since this made it easy to control the learning curves and avoid overfitting. Even using one clean image target, the obtained LIP is practical due to the powerful transferability to other image with no clean groundtruth. Moreover, we can use the noisy target in finding the same effective LIPs as long as we perform

"early-stopping" in the optimization process. Note that it was the same trick mentioned when training the original DIP for denoising tasks (Ulyanov et al., 2018) and was identified to be essential to the original DIP's success in (Heckel & Hand, 2019). In Table. 4, we conduct the comparison experiments of clean and noisy image targets (experimental settings are summarized in the caption of the table) and we find that there are no large differences between the results of noisy image targets and clean image targets.

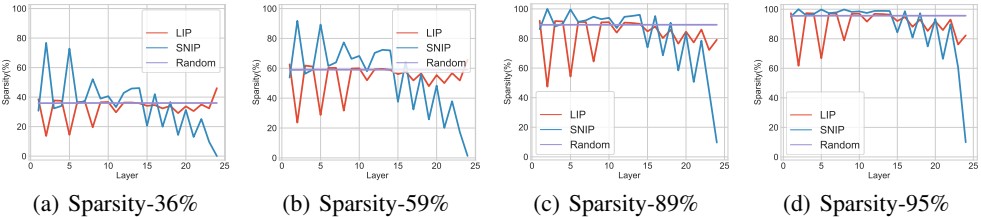

(a) Sparsity-36%  (b) Sparsity-59%  (c) Sparsity-89%  (d) Sparsity-95%

Figure 11: Layer-wise sparsity ratio results of LIP, SNIP and randomly pruned tickets. Note that we summarize the sparsity ratio of each layer: the ratio of the number of parameters whose values are equal to zero to the number of total parameters of the layer. And the x-axis of these figures is composed of the serial numbers of model layers. We sampled subnetworks with four different sparsities (sparsity $= 36\%, 59\%, 89\%, 95\%$) to observe.

**A Closer Look into the Lottery Image Prior Structure**  As shown in Fig. 11, the structure of the LIP subnetwork is drastically different from those found by SNIP and random pruning, in particular the distribution of layerwise sparsity ratios. LIP tends to preserve weights of the earlier layers (closer to the input), while pruning the latter layers more aggressively (e.g, Fig. 11(a)). In contrast, SNIP tends to prune much more of the earlier layers compared to the latter ones. Random pruning by default prunes each layer at approximately the same ratio.

Comparing the three methods seem to suggest that for finding effective and transferable LIP subnetworks, to specifically keep more weights at earlier layers more is important. That is an explainable finding, since for image restoration tasks, the low-level features (color, texture, shape, etc.) presumably matter more and are more transferable, than the high-level features (object categories, etc.). The earlier layers are known to capture more of low-level image features, hence contributing more to retraining the image restoration performance with DIP.

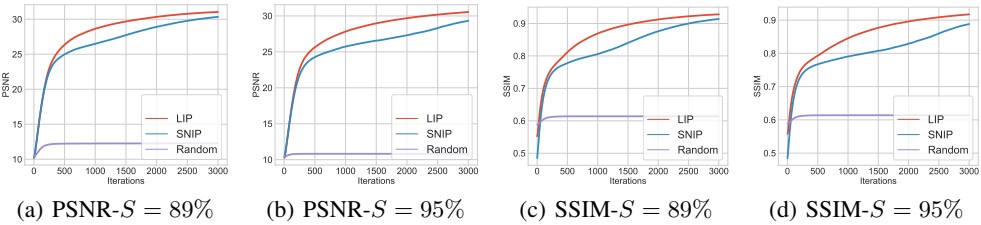

(a) PSNR-$S = 89\%$  (b) PSNR-$S = 95\%$  (c) SSIM-$S = 89\%$  (d) SSIM-$S = 95\%$

Figure 12: The learning curve plots when using different subnetworks towards the DIP task. In the figure, $S$ denotes the sparsity of the model. We compare both PSNR and SSIM values. For fair comparisons, we trained these subnetworks on the denoising task on the Baby image with 3000 iterations, then trained in isolation (the iteration number is recommended by (Chen et al., 2020c) to capture the "early-stopping" phenomenon of DIP), and summarized their performances.

**Learning Curve Comparison of Using Various Subnetworks for the Restoration Task**  We further compare the training convergence curves of different subnetworks on the restoration task. In Fig. 12, we summarize the convergence of LIP, SNIP and randomly pruned subnetworks on the denoising task, and the experimental details are included in the caption. We use the PSNR and SSIM metrics to measure the quality of the generated images: SSIM is often considered better "perceptually aligned", by attending more to the contrast in high-frequency regions than PSNR.

At the early stage of optimization, we observe that the learning curves of LIP and SNIP subnetworks are almost overlapped (either PSNR or SSIM curves), while the randomly pruned subnetworks failed to perform comparably with them. Yet when the iterations increase, the SNIP subnetworks start to lag behind the LIP subnetworks (e.g., the largest PSNR gap between the two can reach 3dB and the largest SSIM gap can be 0.7). Only the LIP subnetworks can match the comparable performances of the full model when reaching the 3000-th iteration. Lastly, the SSIM gap is noticeably enlarged at higher sparsity levels (95%) when comparing LIP and SNIP, which implies LIP to be better at capturing perceptually aligned details.

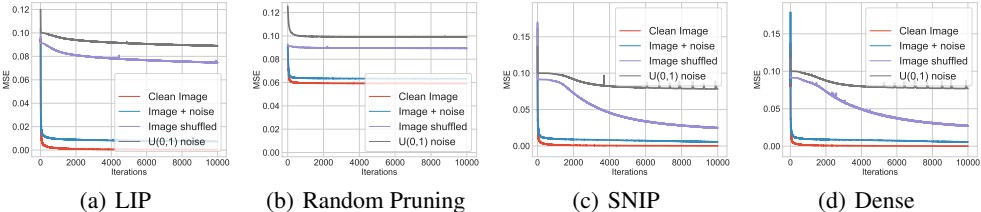

(a) LIP  (b) Random Pruning  (c) SNIP  (d) Dense

Figure 13: Learning curves using four different training targets: a clean image (Baby.png), the same image added with noises, the same randomly scrambled and white noise. Note that we use four different models: the LIP subnetwork ($S = 89\%$), randomly pruned subnetwork ($S = 89\%$), SNIP subnetwork ($S = 89\%$) and the dense model ($S = 0\%$). And we trained them in isolation in the same experimental settings for 10000 iterations.

**Learning Curves of Subnetworks with Different Training Targets**  To better explain the success of the LIP subnetworks, inspired by the Figure 2 of (Ulyanov et al., 2018), we further train the obtained subnetworks (LIP, SNIP and random pruning) with four different targets: 1) a natural image, 2) the same added with noise, 3) the same after randomly permuting the pixels and 4) the white noise. We also train the dense model as the baseline results. The experimental details are summarized in the caption of Fig. 13. We first observe that for LIP, SNIP and dense models, the optimization converges much faster in case 1) and 2) than in case 3) and 4). But the randomly pruned subnetworks have failed in all cases. Interestingly, we also find that SNIP subnetworks perform similarly with the dense model. Meanwhile, the parametrization of LIP subnetworks offers the higher impedance to noise and the lower impedance to signal than the dense model, which indicates that the separation of high frequency and low frequency is more obvious for winning network architectures. Also, in order to observe whether the high-frequency information will be lost during the pruning, we apply Fourier Transformation to the Baby figure (described in Fig. 14) and visualize the frequency intensity of the ground-truth image and the reconstructions from three different subnetworks (LIP, SNIP and random pruning). The results are summarized in Fig. 15. We found that compared with random pruning, LIP and SNIP can maintain most of the high frequency information in the ground-truth (e.g., in Fig 15, SNIP and LIP can both maintain the high-frequency information at the sparsity of 79%; however, SNIP could lose more high-frequency information than LIP at lower sparsity ratios.).

| Sparsity(%) | 0 | 36 | 59 | 67 | 79 | 89 | 95 |
|---|---|---|---|---|---|---|---|
| **Clean Image** | 32.60 | 32.51 | 32.45 | 32.40 | 32.21 | 32.10 | 31.99 |
| **Noisy Image** | 32.60 | 32.56 | 32.46 | 32.41 | 32.22 | 32.13 | 31.94 |

Table 4: We compare the results of clean and noisy image targets in **setting i**. Note that the used image is F16.png, the evaluation metric is PSNR and the training iteration number is 3000 to capture the "early-stopping" phenomenon. The results suggest that there are no large differences (with PSNR value smaller than 0.05) between the performances of subnetworks with clean and noisy image targets (the sparsity ranges from 0% to 95%).

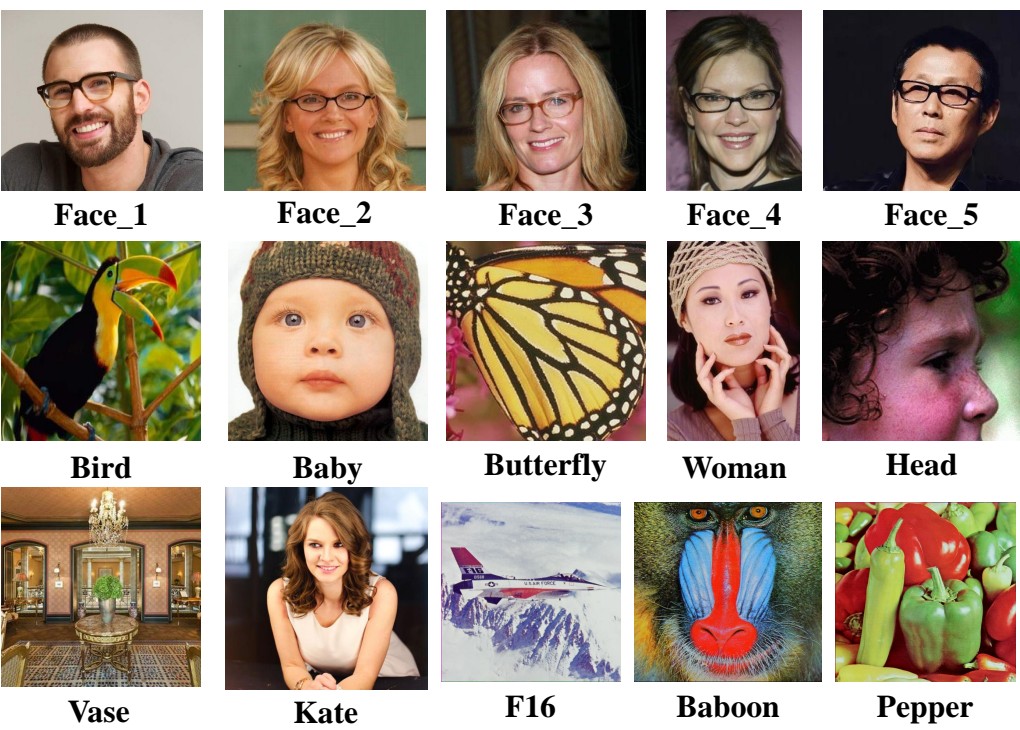

Figure 14: Images used in plotting the curves of experiments.

---

**Algorithm 1:** Single Image-based IMP

---

**Input:** The desired sparsity $s$, the random code $z$ as the model input, the untrained model $f_u$
and the image $x$.

**Output:** A sparse DIP model $f(z; \theta \odot m)$ with image prior property

1 Set $m_u = 1 \in \mathbb{R}^{||\theta||_0}$ and $\theta_0$ are the initial weights of the model $f_u$.

2 Iteration $i = 0$, training epochs $N$ and $j \in [0, N]$;

3 **while** *the sparsity of $m_u < s$* **do**

4      Train the $f_u(z; \theta_0 \odot m_u)$ with the objective $E(f(z; \theta \odot m); x)$ for $N$ epochs to reach the
     parameter $\theta_N^i$;

5      Create the mask $m_u'$;

6      Update the mask $m_u = m_u'$;

7      Set the model parameters to $\theta_j$: $f(z; \theta_j)$;

8      create the sparse model: $f(z; \theta_j \odot m_u)$;

9      $i = i + 1$;

10 **end**

---

---

**Algorithm 2:** Weight-sharing IMP with Various Domain Images

---

**Input:** The desired sparsity $s$, the random code $z$ as the model input, the untrained model $f_u$ and images from $n$ domains $x_a \in \{x_1, x_2, ..., x_n\}$.

**Output:** A sparse DIP model $f(z; \theta \odot m)$ with image prior property

1   Set $m_u = 1 \in \mathbb{R}^{||\theta||_0}$ and $\theta_0$ are the initial weights of the model $f_u$.
2   Iteration $i = 0$, training epochs $N$ and $j \in [0, N]$;
3   **while** *the sparsity of $m_u < s$* **do**
4      loss $= \sum_{a=1}^{n} E(f(z; \theta \odot m); x_a), \ a \in [1, n]$;
5      Train the $f_u(z; \theta_0 \odot m_u)$ by Backpropagation (loss) for $N$ epochs to reach the parameter $\theta_N^i$;
6      Create the mask $m_u'$ and update the mask $m_u = m_u'$;
7      Set the model parameters $f(z; \theta_j)$;
8      create the sparse model $f(z; \theta_j \odot m_u)$;
9      $i = i + 1$;
10   **end**

---

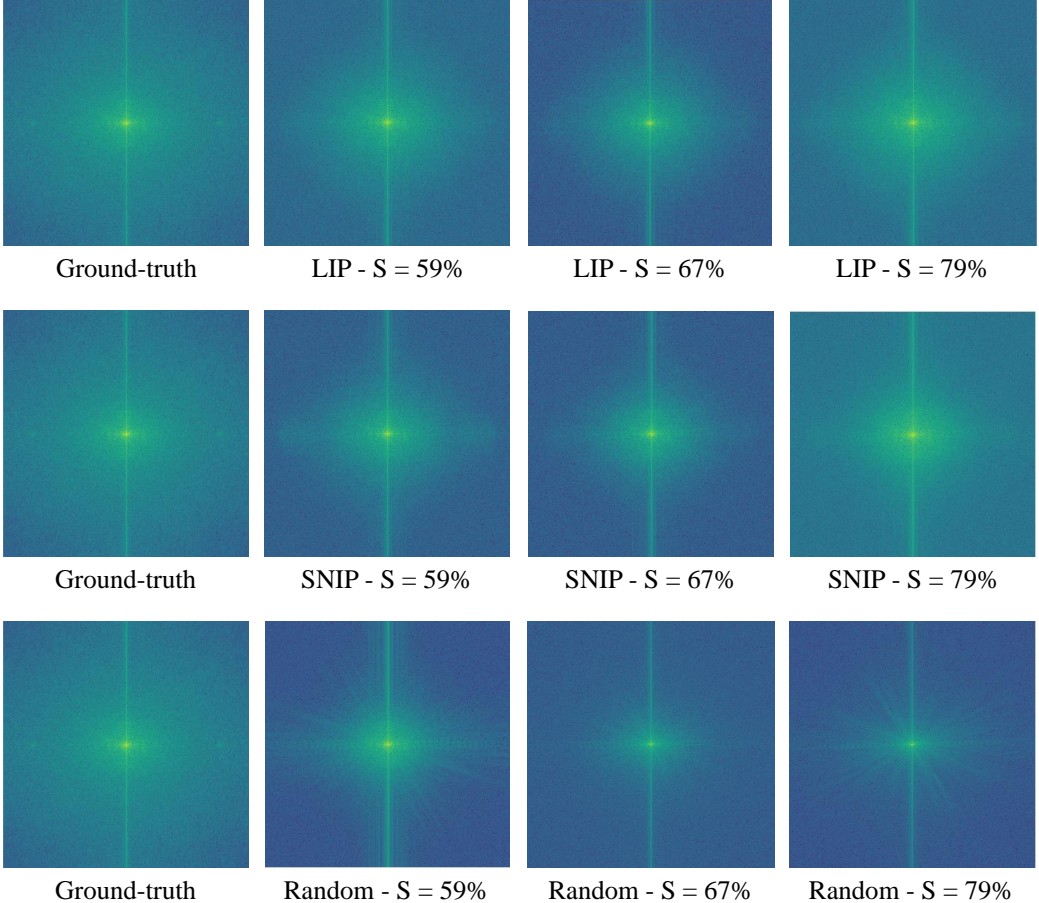

Figure 15: Evaluating the reconstruction images of different subnetworks (LIP, SNIP and random pruning) by FFT (Fast Fourier Transformation) to check whether the high frequency information has been lost during pruning. Note that we experimented on the Baby.png. We found that compared with random pruning, LIP and SNIP can both maintain the high frequency information of the ground-truth. For example, the LIP and SNIP subnetworks both maintain mostof the high-frequency information of the ground-truth at the sparsity 79%, but the LIP could also performs well at the sparsity 59% where the SNIP loses more high-frequency information.

