# OpenReview forum: "Lottery Image Prior"
_ICLR.cc/2022/Conference — ICLR 2022 Submitted_

### Official Review · Reviewer_vf8k · 2021-10-28

**Correctness:** 3
**Technical Novelty And Significance:** 3
**Empirical Novelty And Significance:** 2
**Recommendation:** 6
**Confidence:** 4

**Main Review:**

My questions mainly concentrate on the setting 1 of the paper, i.e., the DIP related tasks.

1. This work shows that there exists a very sparse subnetwork whose performance can surpass the original dense network. To find such sparse network, it directly minimizes the MSE relying on the ground-truth image. In practice, however, the ground-truth image is always not accessible. How to obtain such sparse subnetwork in real applications?

2. In my opinion, the goal of the research on LIP is to simplify the DNN architecture in DIP based on the pre-known knowledge on the sparse subnetwork. For example, given a noisy image without the latent clean image, if we know the sparse subnetwork in advance, we can directly optimize it to obtain the denoising result under DIP.  How to achieve this goal? If LIP cannot embed into current DIP related works like this, it is useless in practice.

3. LIP is implemented by the network weight pruning, and all the experiments depends on a current existing weight pruning method. I think  the contribution of this paper is incremental and not enough.

4. The mathematic notions in algorithm1 and algorithm2 are confused. In algorithm 1, what does $f_{\mu}(x;\theta_0 \odot m_{\mu})$ mean? In DIP, the network $f$ takes the latent code $z$ as input instead of the image $x$. As for the loss function of algorithm 2, whom does $E(f(x_a;\theta \odot m))$ denotes the MSE distance between the network output $f(x_a;\theta \odot m)$ and?

**Summary Of The Paper:**

This paper exploits the LTH in the low-level vision tasks, including image restoration and compressive sensing image reconstruction. Through some experiments, it identifies the existence of LIP in low-level vision and also gives some interesting conclusions on the transferability of the proposed LIP.

**Summary Of The Review:**

Even though this paper is interesting, I still argue the usefulness of the proposed LIP. This paper only attempts to do some exploration based current existing LTH related works, which weakens  the novelties and the contributions of this work.

---

> ### Author Response · Authors · 2021-11-18
> **Rebuttal of Reviewer vf8k**
>
> [Cons1.] Noisy image used as the training target will be more practical.
>
> 1) We used clean images as the training targets since this made it easy to control the learning curves and avoid overfitting. Even using one clean image target, the obtained LIP is practical due to the powerful transferability to other images with no clean ground-truth (discussed in [Cons2.] in detail).
> 2) We can also use the noisy target in finding the same effective LIPs as long as we perform “early-stopping” in the optimization process. Note that it was the same trick mentioned when training the original DIP for denoising tasks [1] and was identified to be essential to the original DIP’s success in [2].
> 3) In fact, our results support both using clean and noisy targets (we added the experiments in Table 4 in the section Supplementary Materials). We actually find there are no large differences between them when sticking to the same appropriate experimental setting.
> 4) Specifically in Table 4, we conducted the IMP experiments on the DIP model with clean and noisy targets on the same image (F16.png, presented in Figure 13). Note that both DIP optimization iteration numbers are 3000 (the number is recommended by the paper “Deep Image Prior” [1] as ``early stopping”) and prune 20% model parameters every IMP epochs. Interestingly, we find there are no large differences (with PSNR value smaller than 0.05) between the performances of subnetworks with clean and noisy image targets (the sparsity ranges from 0% to 95%). Moreover, we add the experiments to explore the convergence curves of subnetworks obtained from different methods. And we also investigate their sparsity distributions per layer. Details are in [Cons2.] Reviewer wMsj.
>
> [1] Ulyanov, Dmitry, Andrea Vedaldi, and Victor Lempitsky. "Deep Image Prior." CVPR, 2018.
>
> [2] Heckel, Reinhard, and Paul Hand. "Deep decoder: Concise Image Representations from Untrained Non-convolutional Networks." ICLR 2019.
>
> [Cons2.] LIP is impractical and it should be directly optimized to obtain the sparse architecture instead of the iterative process.
>
> We respectfully disagree. The found LIP subnetworks by multi-image based IMP are practical because they have the powerful transferability: 1) They can transfer across different images (Figure 5 illustrates this point); 2) They can successfully transfer across different restoration tasks such as denoising, inpainting and super-resolution (Figure 7 illustrates this point). And therefore, there is no need to do the IMP process again and again for obtaining desired LIPs in different tasks. We hope the reviewer will capture this crucial point and give a more fair assessment, instead of rushing to an ungrounded accusation of “useless”.
>
> [Cons3.] Novelty limited?
>
> We respectfully yet firmly disagree with you. The authors are a well-established team in pruning and lottery ticket hypothesis, and we understand the context well. While “iterative weight pruning” is never new, you would miss the point by simply denying our novelty owing to that.
>
> We confidently claim the following important innovations. We hope they would clarify any of your remaining concerns about novelty.
>
> 1) We are the first to demonstrate the existence of LTH in image inverse problems or DNN-based priors. This novelty has been acknowledged by other three reviewers. Reviewer wMsj has acknowledged that “The proposed objective is an important design for the system to work”, “The conclusion is very interesting”. Reviewer Kjf3 has also acknowledged our novelty as “The work deserves merit due to the novel application”, “The paper conveys several new ideas”, “The paper is well written with several questions justified and analyzed thoroughly”. Reviewer jFko claimed that “Originality. This work studies the LTH for low-sample deep inverse imaging for the first time”
>
> 2) We investigated both untrained and pre-trained DNNs as image priors by overcoming several impediments such as severely limited training data and task objective mismatch. This merit has also been acknowledged by other reviewers. Reviewer Kjf3 acknowledged that “The paper addresses an important problem of image restoration using deep prior which has been quite effective in the absence of labeled examples”. Reviewer jFko acknowledged that “Both studying scenarios are dealing with a single image training task, or, an overfitting task, which makes this work different from the past work using LTH.”
>
> 3) Our finding reflects the underlying common image prior that is agnostic to specific data or tasks, through the lens of the CNN architecture together with sparsity. This point has also been praised by other reviewers. Reviewer jFko acknowledged that “the results suggest transferability of LIP subnetworks across data and image restoration tasks”.
>
> [Cons4.] There are confusing mathematic notions in algorithm1 and 2.
>
> Thanks for pointing it out and we have revised the 2 confusing places.

---

### Official Review · Reviewer_jFko · 2021-10-31

**Correctness:** 3
**Technical Novelty And Significance:** 2
**Empirical Novelty And Significance:** 3
**Recommendation:** 6
**Confidence:** 3

**Main Review:**

Originality
This work studies the LTH for low-sample deep inverse imaging for the first time

Significance
Strong points
- for the first time studies LTH for deep image prior and pre-trained deep generative compressed sensing
- provides empirical results for several low-level image restoration (as well as image classification) tasks and datasets
- the results are interesting suggesting LTH exists with high degree of sparsity that not only makes the models much smaller but also improves the generalization performance as a result of sparsity regularization
- the results also suggest transferability of LIP subnetworks across data and image restoration tasks

Weak points
- The paper presentation is poor. There are several typos throughout the text. The organization is poor and confusing. There are several scenarios studied in this paper with regards to the datasets and tasks and settings, but the link among them is missing and confusing from the text.
- This work is purely empirical study and it's not clear for example why LIP can transfer across image restoration tasks, and dataset? Providing insights would be very useful. It is also not quite clear how the LIP pruned for denoising is transferred to image classification. Are only the intermediate representations transferred as in auto-encoders?


More comments and suggestions
- In the modified IMP, when using clean labels for training, where is the randomness coming from? How is the expectation defined?
- The presentation of the paper can benefit from more visuals such as diagrams/graphs to connect the tasks and datasets.
- The experiments only report PSNR. It would be better to report SSIM and FID to see if the pruning can hurt the high frequency perceptual details or not? This is more important for the pretrained GAN model.
- It is not clear how the sparsity pattern looks like for the considered hour-glass network. Isn’t this one of the important points of pruning as advocated in the introduction? A visualization of the sparsity pattern per network layers would be very insightful. It would also be helpful to see if pruning creates a bottleneck in the network and where that bottleneck is.


**Summary Of The Paper:**

Summary
This submission studies lottery ticket hypothesis for deep neural network based inverse imaging. It considers two scenarios: 1-inference for compressed sensing based on pre-trained deep generative models, 2-deep image prior where all network parameters are fit to a single image. Both scenarios are dealing with a single image training task, or, an overfitting task, which makes this work different from the past work using LTH. The empirical results suggest that with a high level of sparsity not only the model size can be made much smaller, but also the generalization can improve which suggests pruning as a regularization. They also suggest good transferability from one image restoration task to another.



**Summary Of The Review:**

This is empirical work studying an important problem. The results are also interesting which suggest a high degree of sparsity for certain learable inverse imaging problems, that closes the gap with the traditional sparsity based inverse imaging algorithms. The presentation of the paper however needs significant improvement to connect the conclusions toward a coherent story.

---

> ### Author Response · Authors · 2021-11-18
> **Rebuttal of Reviewer jFko**
>
> [Cons1.] There are some typos and the logistic link among different tasks, datasets and settings are ambiguous.
>
> Thanks for pointing it out. We have checked and rewritten the full text. Although the implementation process of the GAN CS task and the DIP restoration task is different, yet we combine the two studies in this one paper for two reasons: 1) we are motivated by two current LTH streams: networks with weights trained from scratch [a] and networks with pre-trained weights [b]. 2) Network structures with random weights or pre-trained weights can be priors, which is the commodity we view as the two could be unified. We have added the explanations to Section 3.1-Neural Network as Priors: Two Settings for clearer presentations. Detailed discussions are in the [Cons2.] Reviewer wMsj.
>
> [a] Frankle, Jonathan, and Michael Carbin. "The lottery ticket hypothesis: Finding sparse, trainable neural networks." ICLR 2019
>
> [b] Chen, Tianlong, et al. "The lottery tickets hypothesis for supervised and self-supervised pre-training in computer vision models." CVPR 2021
>
> [Cons2.] Try to explain the success of transferability of LIP (studying the sparsity pattern per network layer will help) and how to transfer DIP models to image classifications.
>
> We have added the results of layer-wise sparsity ratio of different subnetworks (LIP, SNIP, Randomly Pruned) in Figure 11 and also plotted the convergence curves in Figure 13 of these subnetworks to better explore the reasons for the powerful transferability of LIP subnetworks. We have found that LIP tends to preserve weights of the earlier layers (closer to the input), while pruning the latter layers more aggressively (e.g, Figure 11(a).). In contrast, SNIP tends to prune much more of the earlier layers compared to the latter ones. Random pruning by default prunes each layer at approximately the same ratio. Comparing the three methods seems to suggest that to find effective and transferable LIP subnetworks, specifically keeping more weights at earlier layers more is important. That is an explainable finding, since for image restoration tasks, the low-level features (color, texture, shape, etc.) presumably matter more and are more transferable,  than the high-level features (object categories, etc.). The earlier layers are known to capture more low-level image features, hence contributing more to retraining the image restoration performance with DIP. The detailed discussions are included in [Cons1.] Reviewer wMsj.
>
> Regarding “transfer DIP models to image classification”, we modify the model structure to transfer it from restoration tasks to the image classification. We replace the last convolutional layer with the linear layer (with Pytorch framework) and train this new model (with the same initial weight on the restoration tasks) on the CIFAR10 and MNIST dataset from scratch (we have added the experimental details in the caption of Figure 8 in the paper). The failure of transferring LIP subnetwork to image classification could also be partially explained by the aforementioned sparsity distribution discrepancy: LIP subnetwork tends to prune more weights from later layers, which might damage the network’s capability in capturing semantic features (usually needing mid-to-high network layers).
>
> [Cons3.] In the modified IMP, when using clean labels for training, where is the randomness coming from? How is the expectation defined?
>
> The randomness only comes from the input random tensor z (defined in Equation 1). In the single-image based IMP, there are no expectations. While in the multi-image based IMP, the expectation is the average loss value of multiple images.
>
> [Cons4.] Provide more evaluation results by using SSIM and FID to check if the pruning will hurt the high-frequency information of the generated images.
>
> Thanks for pointing it out. We have added the results in Figure 12 by evaluating the generated images with SSIM standards. In Figure 12, we summarize the convergence of LIP, SNIP and randomly pruned subnetworks on the denoising task, and the experimental details are included in the caption. We use the PSNR and SSIM metrics to measure the quality of the generated images: SSIM is often considered better ``perceptually aligned", by attending more to the contrast in high-frequency regions than PSNR.
>
> At the early stage of optimization, we observe that the learning curves of LIP and SNIP subnetworks are almost overlapped (either PSNR or SSIM curves), while the randomly pruned subnetworks failed. Yet when the iterations increase, the SNIP subnetworks start to lag behind the LIP subnetworks (e.g., the largest PSNR gap between the two can reach 3dB and the largest SSIM gap can be 0.7). Only the LIP subnetworks can match the comparable performances of the full model when reaching the 3000-th iteration. Lastly, the SSIM gap is noticeably enlarged at higher sparsity levels (95%) when comparing LIP and SNIP, which implies LIP to be better at capturing perceptually aligned details.

---

> ### Author Response · Authors · 2021-11-23
> **A Kind Reminder to Reviewer jFko**
>
> Dear Reviewer jFko,
>
> We thank you for your valuable time spent reviewing our work, and we really hope to have a further discussion with you to see if our response resolved your concerns. Within the rebuttal period, we have made the following improvements based on your valuable and constructive suggestions, which have been integrated into the latest version of pdf:
>
> 1. We added more explanation to Section 3.1-”Neural Network as Priors: Two Settings” for a stronger logistic link between the two settings considered in this work, DIP and GAN CS, and a clearer presentation.
>
> 2. In an attempt to draw more insights into the powerful transferability of LIP, we made empirical observations on the layer-wise sparsity ratios of different subnetworks (LIP, SNIP, Randomly Pruned) in Figure 11 and their convergence curves in Figure 13. We found that it is more important to keep more weights at earlier layers for effective and transferable LIP subnetworks.
>
> 3. We reported the SSIM results of LIP and other baselines in Figure 12, we found that LIP is better at capturing perceptually aligned details than SNIP and random pruning subnetworks.
>
> 4. We clarified the ambiguity in the original manuscript, such as the source of randomness when using clean labels for IMP and how to transfer to image classification. Thank you again for pointing them out, which definitely helped to make our write-up better.
>
> We would sincerely appreciate it if you could kindly share your thoughts on the key points in our response, and keep the discussion rolling in case you have further comments. Thank you!
>
> Best wishes,
>
> Authors

---

> > ### Comment · Reviewer_jFko · 2021-11-23
> >
> > The authors have put effort in addressing my comments. The pruning is more explainable now. The reason for pointing out SSIM was to make sure high frequency information is not lost significantly with the pruning. SSIM is still very correlated with PSNR and not the best metric to assess the high frequencies. A visualization of Fourier domain histogram could be more useful to be included in the final paper. About the classification, I don't quite agree with the author's arguments about lack of transferability. Typically bottleneck layers with presumably higher levels of pruning/sparsity carry more contextual information and would lead to better downstream classification. Also, one could always finetune on top of the intermediate layers if more level information is needed for the classification task. I would recommend the authors to clarify that and not to make general conclusions out of this observation. Overall, I appreciate the authors efforts and even though this is a purely empirical study I find the outcomes interesting. Hence, I would raise my score.

---

> > > ### Author Response · Authors · 2021-11-23
> > > **Response to the Comments**
> > >
> > > Thanks for your constructive feedback and positive re-evaluation of our work.
> > >
> > > 1. We added visualization results on the frequency intensities of the Fourier transformation of the Baby image from Set-5 (Figure 15 at the end of the Supplementary). Specifically, we performed the visualization for the ground-truth image and the reconstruction images of LIP, SNIP and Random Pruning with different levels of sparsity. We observed that compared with random pruning, LIP and SNIP can maintain most of the high frequency information of the ground-truth. The detailed discussions are included in section - Supplementary Materials - Learning Curves of Subnetworks with Different Training Targets.
> > >
> > > 2. We agree with your comments on the potential transferability to classifications. We will tone down the "conclusions" we previously drew in the discussion and focus more on describing only our observations. Meanwhile, we will explore the idea you suggested: add and finetune linear layers on top of the intermediate layers, i.e., transfer only the winning tickets of the encoder.
> > >
> > > Best,
> > >
> > > Authors

---

### Official Review · Reviewer_Kjf3 · 2021-11-02

**Correctness:** 3
**Technical Novelty And Significance:** 3
**Empirical Novelty And Significance:** 3
**Recommendation:** 8
**Confidence:** 5

**Main Review:**

Positives

+ The paper addresses an important problem of image restoration using deep prior which has been quite effective in the absence of labelled examples.

+ The work deserves merit due to the novel application of the lottery ticket hypothesis to this problem and has been well presented.

+ The paper conveys several new ideas and is shown to induce sparsity on the network while achieving impressive results.

+ The paper is well written with several questions justified and analysed thoroughly.

Concerns

- The paper does have a lot of typos such as the line - "that can be training in isolation" in two different places.

- Experimental evaluation and comparison for deep image prior tasks lacking rigour.

- The paper does not cite several works on line of the deep image prior such as those listed below. they have not been compared with.

1. Gandelsman, Y., Shocher, A., & Irani, M. (2019). " Double-DIP": Unsupervised Image Decomposition via Coupled Deep-Image-Priors. In Proceedings of the IEEE/CVF Conference on Computer Vision and Pattern Recognition (pp. 11026-11035).
2. I. D. Mastan and S. Raman, "DCIL: Deep Contextual Internal Learning for Image Restoration and Image Retargeting," 2020 IEEE Winter Conference on Applications of Computer Vision (WACV), 2020, pp. 2355-2364
3. Mastan, I. D., & Raman, S. (2021). DeepCFL: Deep contextual features learning from a single image. In Proceedings of the IEEE/CVF Winter Conference on Applications of Computer Vision (pp. 2897-2906).



**Summary Of The Paper:**

The paper presents the application of the Lottery Ticket Hypothesis (LTH) to the problem of the deep image prior in order to solve image restoration tasks. The paper shows the effectiveness of such a lottery image prior which trains subnets in isolation.

**Summary Of The Review:**

The paper has several positives, the concerns include missing references and comparisons. Hence, it is just below acceptance threshold.

---

> ### Author Response · Authors · 2021-11-18
> **Rebuttal of Reviewer Kjf3**
>
> [Cons1.] The paper does have some typos.
>
> Thanks for pointing it out. We have checked and rewritten the full paper.
>
> [Cons2.] Experimental evaluation and comparison for deep image prior tasks lacking rigor.
>
> We respectfully disagree. Firstly, in the paper “Playing the lottery with rewards and multiple languages: lottery tickets in RL and NLP” [1] (ICLR 2020), the authors study the lottery ticket hypothesis problem in NLP and Reinforcement Learning fields. While they only compare the LTH IMP with the random pruning, we also compared the LTH IMP with the SNIP method, which better clarifies the effectiveness of our proposed LIP finding approach. Moreover, we studied the specific rewind strategy in our experiments with the 0%, 5%, 10%, 20% rewinding weights just as the paper “GANs Can Play Lottery Tickets Too” [2] (ICLR 2021) did to study the problem with caution. Last but not least, we have added the experiments of randomly pruned GAN priors in compressed sensing and inpainting tasks to demonstrate that it is not easy to find the winning LIPs in GAN structures. Apart from that, we have added the layer-wise sparsity ratio experiments and the convergence curves of different subnetworks to explore possible reasons for the success of LIP subnetworks. Therefore, we are confident to claim that our experimental evaluations are thorough and can well support our conclusions.
>
> [1] Yu, Haonan, et al. "Playing the lottery with rewards and multiple languages: lottery tickets in rl and nlp." ICLR 2020.
>
> [2] Chen, Xuxi, et al. "Gans can play lottery tickets too." ICLR 2021.
>
> [Cons3.] Lack citation of following papers relevant to DIP and lack comparisons with them.
>
> Thanks for pointing it out! All mentioned works are interesting, and we have cited and discussed these relevant papers in the Background Work Section.
>
> Regarding “lack of comparison with these works”, we were not so sure since these mentioned works aim to create various image prior or internal learning techniques for the restoration tasks, which is completely orthogonal to our goal of discovering LTH subnetwork from the classical DIP.
>
> We hope you kindly agree that it is unnecessary and hardly meaningful to seek any direct apple-to-apple comparison with them, due to their vastly different research goals. We will be happy to explore whether similar LTH phenomena can be observed in their variants of DIP in future work.

---

### Official Review · Reviewer_wMsj · 2021-11-03

**Correctness:** 3
**Technical Novelty And Significance:** 3
**Empirical Novelty And Significance:** 3
**Recommendation:** 6
**Confidence:** 4

**Main Review:**

As I have mentioned in the above summary, the conclusion (some networks are more suitable to be used as deep image prior than others) is very interesting. It seems that the author only discussed the existence of these networks, but did not discuss the possible reasons why these structures is superior to other structures in the deep image prior task. Is this because in these networks, the important information (such as low-frequency parts, etc.) of the image used for recovery is easier to learn from the obtained network structure, while the high-frequency, harmful information is more difficult to capture by these networks? The lack of this part of the discussion is regrettable. It is possible that for DIP, the preferred network structure is different from discriminative tasks and GANs. I hope the author can introduce relevant discussions. If possible, some experiments can also be conducted. For example, maybe the convergence curves of DIP are different when different networks are used, and the separation of high frequency and low frequency is more obvious for winning network architectures?

Another regret is that GAN compressed sensing and DIP seem to be separated. It is not clear whether the cause of the effect on GAN CS and the result on DIP are the same. It is very likely that the reason for the effective pruning on GAN CS is different from DIP. Maybe the former is because some knowledge in GAN has no or negative influence on the CS task, while the latter is because different structures express different components of the image differently. After reading this paper, I still lack knowledge of these issues.

Another minor problem is that this paper is difficult to follow. The writing needs improvement. Especially in terms of specific methods, I think it is difficult to quickly understand the author's approach. Moreover, the practices of GAN CS and DIP are very different, and it is difficult to put them together clearly.

**Summary Of The Paper:**

This paper researches the lottery ticket hypothesis for networks as a deep image prior or deep generative prior. The specific approach is to (1) train deep networks to reconstruct multiple images for DIP (Ticket ﬁnding objectives), (2) conduct iterative magnitude pruning to the trained network, (3) obtain the pruned mask and reset the model parameter to initialization weight and (4) perform deep image prior to new (or training) images.
The proposed objective is an important design for the system to work, i.e., the author optimizes the network by minimizing the expected error on multiple images.
The experimental results are somehow interesting that (1) different pruning methods actually performs differently and (2) there are winning tickets in the studied problem. As for GAN compressive sensing task, there are also winning tickets that exists. The point I am more interested in is that some networks are more suitable to be used as deep image prior than others.

**Summary Of The Review:**

Overall, this is an interesting paper. This paper reveals some interesting results, but there are still some questions (see above). If the author can discuss these issues and improve writing, the paper can be accepted.

---

> ### Author Response · Authors · 2021-11-18
> **Rebuttal of Reviewer wMsj**
>
> We are very glad you had a positive initial impression and we provide pointwise responses for your concerns below.
>
> [Cons1.] Why the specific lottery image prior structure is superior to other structures in DIP tasks?
>
> Thanks for the advice. We add the experiments of studying the layer-wise sparsity ratio and the learning curves of different subnetworks, to add to the interpretability of what is special about LIP subnetworks. The results are summarized in Figure 11 and 13 in the section - Supplementary Materials of the paper.
>
> As shown in Figure 11, the structure of the LIP subnetwork is drastically different from those found by SNIP and random pruning, in particular the distribution of layerwise sparsity ratios. LIP tends to preserve weights of the earlier layers (closer to the input), while pruning the latter layers more aggressively (e.g, Figure 11(a).). In contrast, SNIP tends to prune much more of the earlier layers compared to the latter ones. Random pruning by default prunes each layer at approximately the same ratio.
>
> Comparing the three methods seems to suggest that for finding effective and transferable LIP subnetworks, specifically keeping more weights at earlier layers more is important. This is an explainable finding because, in image restoration tasks, the low-level features (color, texture, shape, etc.) presumably matter more and are more transferable,  than the high-level features (object categories, etc.). The earlier layers are known to capture more low-level image features, hence contributing more to retraining the image restoration performance with DIP.
>
> To better explain the success of the LIP subnetworks, inspired by the Figure 2 of the paper “Deep Image Prior”, we further train the obtained subnetworks (LIP, SNIP and random pruning) with four different targets: 1) a natural image, 2) the same added with noise, 3) the same after randomly permuting the pixels and 4) the white noise. We also train the dense model as the baseline. The experimental results are summarized in Figure 13. We first observe that for LIP, SNIP and dense models, the optimization converges much faster in case 1) and 2) than in case 3) and 4). But the randomly pruned subnetworks have failed in all cases. Interestingly, we also find that SNIP subnetworks perform similarly with the dense model. Meanwhile, the parametrization of LIP subnetworks offers a higher impedance to noise and a lower impedance to signal than the dense model, which indicates that the separation of high frequency and low frequency is more obvious for winning network architectures.
>
> The failure of transferring LIP subnetwork to image classification could also be partially explained by the aforementioned sparsity distribution discrepancy: LIP subnetwork tends to prune more weights from later layers, which might damage the network’s capability in capturing semantic features (usually needing mid-to-high network layers).
>
>
> [Cons2.] DIP task and GAN CS task seem to be separated. The reasons for the success of GAN tickets on CS and DIP tickets on restoration tasks are different.
>
> You are correct that the implementation process of the GAN CS task and the DIP restoration task is different, yet we combine the two studies in this one paper for two reasons: 1) we are motivated by two current LTH streams: networks with weights trained from scratch [a] and networks with pre-trained weights [b]. 2) Network structures with random weights or pre-trained weights can be priors, which is the commodity we view as the two could be unified.
>
> Also, in setting i, we have added the experiments of analyzing the layer-wise sparsity ratio and the convergence curves of different subnetworks. In setting ii, we have added the randomly pruned PGAN as the comparison results (the experiments are running, we will update the results later). We have added the explanations to Section 3.1-Neural Network as Priors: Two Settings for clearer presentations.
>
> We have detailedly discussed the inner structure of the LIP subnetworks in [Cons1.] and we find that keeping more weights at earlier layers more is important for finding effective and transferable LIP subnetworks. We also observed that the separation of high frequency and low frequency is more obvious for winning network architectures based on the learning curves of various subnetworks.
>
> [a] Frankle, Jonathan, and Michael Carbin. "The lottery ticket hypothesis: Finding sparse, trainable neural networks." ICLR 2019
>
> [b] Chen, Tianlong, et al. "The lottery tickets hypothesis for supervised and self-supervised pre-training in computer vision models." CVPR 2021
>
>
> [Cons3.] Need improving the presentation of the specific method and the two experimental parts: GAN CS and DIP.
>
> Thanks for your advice on the writing. The logistic link between the GAN CS and DIP is detailedly discussed in [Cons2.] and we have added the above explanations to the Section 3.1-Neural Network as Priors: Two Settings.

---

> > ### Comment · Reviewer_wMsj · 2021-11-19
> > **Reply to the Rebuttal**
> >
> > I have read the author feedback. Glad to see the new discussions. I noticed that the writing has been revised. Thus I keep my positive rating.

---

### Decision · Program_Chairs · 2022-01-20

**Decision:**

Reject

**Comment:**

The paper studies the lottery ticket hypothesis in the context of deep image priors. Deep image priors are convolutional neural networks that are imposed as a prior for image reconstruction problems. A deep image prior can be an un-trained convolutional network, or it can be a trained generator, and the paper considers both types of priors. Deep image priors are often highly over-parameterized and thus the paper under review asks the question on whether the networks really have to be heavily parameterized or whether a small subnetwork will also do. The paper performs experiments on image restoration with an entirely un-trained DNN (this constitutes the larges part of the paper) and on image restoration with a pre-trained network.

The paper received four reviews out of which three recommend weak acceptance and one strong acceptance.
- Reviewer wMsj finds it interesting that the paper shows that some networks are more suitable as deep image priors than others, but finds that the paper lacks evidence on why some structures are better than others.
- Reviewer Kjf3 strongly recommends acceptance (8) in a relatively generic review. The reviewer finds the paper is interesting as it provides a novel application of the lottery ticket hypothesis. However, the review also criticizes that the experimental evaluating lacks rigor.
- Reviewer jFko appreciates that the paper studies the lottery ticket hypothesis for the first time for un-trained and pre-trained image prior, and that the results are interesting as they suggest that the models can be made smaller and that this can even improve performance. The reviewer criticizes that the paper's presentation is confusing, and points out a few weaknesses in the empirical evaluation. The authors responded and after a brief discussion, the reviewer raised their score.
- Reviewer vf8k provides a relatively brief review and criticizes that to find sparse networks, one requires the ground truth image, which is not accessible. The authors clarify that the method is transferable in that the network identified can be used for other images and is thus transferable. The reviewer was satisfied with this response.

The score of this paper 6.5 after the discussion period. Three of the reviewers are on the fence about the paper, one reviewer is not, and that reviewer significantly impacted the score. This reviewer, however, did not provide convincing arguments about the merits of the paper. All reviewers find that `3: Some of the paper’s claims have minor issues.', and I agree with that statement.

I do not recommend acceptance of the paper in its current form, because of insufficiently rigorous experiments to justify the claims:
- Specifically, the paper's goal is to address the research question 'do they [neural-network based priors for image reconstruction] really have to be heavily parameterized?' The literature already answers this question since as the paper under review reads on page 3, the literature found that an under-parameterized non-convolutional model can function as an un-trained image prior. Those underparameterized networks perform well and have fewer parameters than the best-performing networks found in the paper under review.
- The paper argues that a sparse network can give better performance. This claim is based on an at most 0.1dB difference, which can only be achieved when choosing the optimal sparsity level, which is not clear to do without knowing the ground truth image.